# Nanopore Sequencing Allows Recovery of High-Quality Completely Closed Genomes of All *Cronobacter* Species from Powdered Infant Formula Overnight Enrichments

**DOI:** 10.3390/microorganisms12122389

**Published:** 2024-11-22

**Authors:** Narjol Gonzalez-Escalona, Hee Jin Kwon, Yi Chen

**Affiliations:** 1Genomics Development and Applications Branch, Division of Food Safety Genomics, Office of Applied Microbiology and Technology (OAMT), Office of Laboratory Operations and Applied Science (OLOAS), Human Foods Program, Food & Drug Administration, College Park, MD 20740, USA; 2Microbial Methods Development Branch, Division of Food and Environmental Safety, Office of Applied Microbiology and Technology (OAMT), Office of Laboratory Operations and Applied Science (OLOAS), Human Foods Program, Food & Drug Administration, College Park, MD 20740, USA; heejin.kwon@fda.hhs.gov (H.J.K.); yi.chen@fda.hhs.gov (Y.C.)

**Keywords:** nanopore sequencing, foodborne pathogen, complete genomes, long read sequencing, *Cronobacter*, powdered infant formula

## Abstract

Precision metagenomic approaches using Oxford Nanopore Technology (ONT) sequencing has been shown to allow recovery of complete genomes of foodborne bacteria from overnight enrichments of agricultural waters. This study tests the applicability of a similar approach for *Cronobacter* genome recovery from powdered infant formula (PIF) overnight enrichments, where *Cronobacter* typically dominates the overall microbiome (>90%). This study aimed to test whether using ONT sequencing of overnight PIF enrichments could recover a completely closed *Cronobacter* genome for further genomic characterization. Ten PIF samples, each inoculated with different *Cronobacter* strains, covering *Cronobacter sakazakii*, *C. muytjensii*, *C. dublinensis*, *C. turicensis*, and *C. universalis*, were processed according to the Bacteriological Analytical Manual (BAM) protocol. Real-time quantitative PCR (qPCR) was used for initial screening (detection and quantification) of the overnight enrichments and confirmed that the inoculated PIF samples after the overnight enrichment had high levels of *Cronobacter* (10^7^ to 10^9^ CFU/mL). DNA from overnight PIF enrichments was extracted from the enrichment broth and sequenced using ONT. Results showed that ONT sequencing could accurately identify, characterize, and close the genomes of *Cronobacter* strains from overnight PIF enrichments in 3 days, much faster than the nearly 2 weeks required by the current BAM method. Complete genome recovery and species differentiation were achieved. This suggests that combining qPCR with ONT sequencing provides a rapid, cost-effective alternative for detecting and characterizing *Cronobacter* in PIF, enabling timely corrective actions during outbreaks.

## 1. Introduction

Precision metagenomics is an approach that customizes the analysis of a metagenomic sample from any matrix to detect and classify a specific pathogen [1]. Developing culture-independent methods for detecting foodborne pathogens can expedite source tracking and reduce the time to implement corrective measures during suspected case scenarios [2,3,4,5,6,7]. For qualitative analysis of foodborne pathogens, qPCR or metagenomic detection is sufficient to call presumptive positive results, which are then followed by microbiological isolate confirmation and potentially subsequent regulatory actions [7]. A similar scenario applies to *Cronobacter* [6,7,8]. The presence of *Cronobacter* in powdered infant formula (PIF) above zero in 300 g is deemed adulterated according to the Code of Federal Regulations of USA (https://www.ecfr.gov/current/title-21/chapter-I/subchapter-B/part-106/subpart-B/section-106.55, accessed on 15 November 2024). 

*Cronobacter* in powdered infant formula (PIF) has been associated with infant illnesses and human clinical cases [8,9,10]. *Cronobacter* was previously classified as one single species, *Enterobacter sakazakii*. Since 2008, it has been reclassified as a genus with seven identified species [11,12]. Among these species, *Cronobacter sakazakii* belonging to ST4 is responsible for over 90% of infant illnesses [13]. Potential sources of PIF contamination with *Cronobacter* spp. during commercial manufacturing include ingredients added after pasteurization, and equipment contamination post-pasteurization, such as spray dryers and fillers [14]. Current *Cronobacter* spp. detection protocols as outlined by the FDA Bacteriological Analytical Manual (BAM) Chapter 29 [15] employs enrichment in buffered peptone water (BPW), qPCR screening, chromogenic agars for isolation, and cultural confirmation by qPCR or biochemical assays.

However, during outbreak investigations, *Cronobacter* spp. detection from PIF is not enough and a complete genome characterization of the *Cronobacter* spp. present is paramount to provide a correct analysis and to check if the isolated strain matches the current outbreak strain. The FDA BAM method does not identify individual species of *Cronobacter*. Single-colony isolation is used for whole genome sequencing (WGS), which confirms the genus identity and determines the species identity and relationship to other *Cronobacter* strains in the WGS database or if it matches a current outbreak strain. This entire process can take approximately 2 weeks of analysis time (Figure 1).

To expedite the analysis time, we aim to investigate Culture-Independent Diagnostics Tests (CIDT) using sequencing methods for the detection and classification of *Cronobacter* strains directly from PIF enrichments. This can be accomplished by using short or long-read sequencing. Bertrand et al. (2019) [16] outlined the challenges associated with short-read sequencing, particularly in accurately assembling complex, highly repetitive genomic regions, especially in multi-species environments. They noted that clustering-based species binning lacks precision for strain-level metagenomic assemblies crucial for outbreak investigations. In contrast, Oxford Nanopore Technologies (ONT) (Oxford, UK) long-read sequencing provides closed genomes while also providing an affordable and portable platform [17]. Current bioinformatics tools, such as EPI2ME or EPI2ME labs (https://labs.epi2me.io/, accessed on 15 November 2024) from Oxford Nanopore Technologies, Galaxy (https://usegalaxy.org/) [18], Bugseq (https://bugseq.com/) [19], MUFFIN (https://github.com/RVanDamme/MUFFIN) [20], and GalaxyTrakr (https://galaxytrakr.org/) [21], offer user-friendly, cloud-based platforms that streamline metagenomic analysis for pathogen detection and characterization. These tools enable non-bioinformaticians in food safety laboratories to perform accurate taxonomic classification and genomic assembly with minimal computational training, thus facilitating the implementation of metagenomic workflows in routine surveillance and outbreak investigations.

The limits of assembly of the long-read nanopore sequencing for use as a precision metagenomics technique from enrichments was established previously as requiring a minimum of 10^6^ CFU/mL of overnight enrichment [1]. The main difference between the *E. coli* precision metagenomic approach from enriched agricultural water and the *Cronobacter* from PIF is that *E. coli* levels in enriched agricultural samples can vary greatly, requiring a certain threshold for successful ONT sequencing to recover a complete genome for further characterization. In the case of *Cronobacter* in PIF, it has been previously demonstrated that the levels are usually between 10^7^ and 10^9^ CFU/mL in overnight enrichments (an internal study). During the enrichment process, *Cronobacter* spp. concentration increases to high levels and can be easily quantified by qPCR, according to the BAM Chapter 29 protocol [15]. Therefore, the levels are high enough in the enrichment for the sample to be considered almost as a pure cultured strain. 

Despite the benefits of nanopore sequencing, including affordability, portability, long reads, and real-time basecalling, the inherent error rate of nanopore sequencing (when using the fast-calling model with the R9.4.1 flow cell and related library preparation kits) precludes the closed genomes from being used for phylogenic analysis [22]. However, in late 2022 ONT released the R10.4.1 flow cell and the rapid barcoding Kit 24 V14 (RBK-114.24), which, together with a newer basecalling model, showed an increase in accuracy (~Q40) [23,24] (https://rrwick.github.io/2023/12/18/ont-only-accuracy-update.html, accessed on 15 November 2024). Additionally, the ONT protocols have improved to allow successful sequence processing of up to 24 samples (genomes of median ~4.5 Mb) in a single flow cell and still obtain a complete closed bacterial genome with a minimum 40X coverage that have high accuracy (>99.9%) and minimal single nucleotide polymorphism (SNP) differences from the reference genome, depending on the species (internal study data not published) [24,25,26]. Consequently, this pilot study aims to test whether we can characterize and close the genomes of up to 10 *Cronobacter* spp. strains that were inoculated into PIF and enriched overnight in a single ONT sequencing run.

## 2. Materials and Methods

### 2.1. Bacterial Strains 

The bacterial strains used in this study belonged to five different *Cronobacter* species (Table 1) that were previously confirmed as *Cronobacter* by qPCR and/or whole genome sequencing. They were selected because they represent a broad diversity of the *Cronobacter* genus. Bacterial strains were cultured in BD Difco™ brain heart infusion (BHI) broth (Thermo Fisher Scientific, Waltham, MA, USA) and stored in BHI broth with 20% glycerol stocks at −80 °C. For the inoculation step the strains were grown in BHI broth at 36 °C for 24 h and then 10-fold diluted in phosphate-buffered saline (PBS) and added to 25 g portions of PIF sample as described below.

### 2.2. Artificial Contamination and Sample Processing 

Powdered cow milk-based infant formula (PIF) sample was obtained locally and tested negative for *Cronobacter* prior to use. For artificial contamination, individual 25 g portions of the PIF sample were inoculated separately with 10 to 15 CFU of each *Cronobacter* strain and mixed with 225 mL of buffered peptone water (BPW) (Thermo Fisher Scientific). The samples were then incubated at 36 °C for 24 h. An aliquot of each overnight enrichment was processed according to the procedure in Chapter 29 of the BAM, as outlined in Figure 1. An additional aliquot of 1 mL was set aside for further processing with Nanopore sequencing. 

### 2.3. DNA Extraction and Cronobacter qPCR Detection Following the BAM Procedure 

Briefly, after BPW enrichment, 40 mL of enrichment cultures were centrifuged at 3000× *g* for 10 min. The resultant pellet was washed and resuspended in PrepMan Ultra (Thermo Fisher Scientific, Waltham, MA, USA) and boiled for 5 min. The supernatant was used for qPCR in BAM Chapter 29, which targets the *Cronobacter* partial macromolecular synthesis operon: the ribosomal protein S21 (rpsU) gene 3′ end and the DNA primase (dnaG) gene 5′ end [27]. Briefly, two µL of the DNA extract was added to 23 µL master mix containing 0.4 µM *Cronobacter* forward and reverse primers, 0.15 µM of Internal amplification control (IAC) [28], forward and reverse primers, 0.3 µM of *Cronobacter* probe, 0.15 µM of an in-house-developed IAC probe described in Chapter 29 of the BAM, 1X iQ qPCR Supermix (BIORAD, Hercules, CA, USA), additional 2.5 U of Taq polymerase and 3 mM of MgCl_2_ and 50 nM of ROX (Thermo Fisher Scientific) passive dye. All primers and probes (Appendix A) employed in this study were purchased from IDT (Coralville, IA, USA). qPCR was performed on an ABI 7500 Fast Thermal Cycler (software version 2.3, Thermo Fisher Scientific). The qPCR conditions were an initial denaturation at 95 °C for 3 min, followed by 40 cycles of 95 °C for 15 s, annealing at 52 °C for 40 s and extension at 72 °C for 15 s. For each qPCR run we used DNA from pure *Cronobacter* culture as positive control and water as negative control. The level of *Cronobacter* in the enrichment cultures was determined based on qPCR Cq values and an internally established standard curve.

### 2.4. DNA Extraction and Whole Genome Sequencing and Assembly

An aliquot of 1 mL from each PIF inoculated overnight enrichment sample was used for DNA extraction. Genomic DNA from each PIF inoculated sample was extracted using the Maxwell RSC Cultured Cells DNA kit with a Maxwell RSC Instrument (Promega Corporation, Madison, WI, USA) according to the manufacturer’s instructions for Gram-negative bacteria with additional RNase treatment. DNA concentration was determined by Qubit 4 Fluorometer and the dsDNA Quantitation High Sensitivity assay kit (Invitrogen, Carlsbad, CA, USA) according to the manufacturer’s instructions. 

DNA recovered from PIF inoculated enrichment samples were sequenced using a GridION nanopore sequencer (Oxford Nanopore Technologies, Oxford, UK). The sequencing libraries were prepared using the Rapid Barcoding Kit 24 V14 (SQK-RBK114.24) and run in FLO-MIN114 (R10.4.1) flow cells, according to the manufacturer’s instructions, for 48 h. There were only two deviations from the original SQK-RBK114.24 library protocol: (1) the starting DNA per samples was 200 ng, and (2) the barcoding was done using 18 µL of sample DNA plus 2 µL of each barcode. The run was live basecalled using Guppy v 7.1.4 included in the MinKNOW v 23.07.12 software using the super-accurate basecalling model. The initial classification of the reads for each run was done using the “What’s in my pot” (WIMP) workflow contained in the EPI2ME cloud service. That workflow allows for taxonomic classification of the reads generated by the GridION sequencing in real time. The genomes for each inoculated sample were obtained by de novo assembly using all nanopore data output per sample using the Flye program v2.9 [29], using the following parameters: --nano-hq, --read-error 0.03, and -i 4. The assembled contigs were classified by taxonomy by Kraken 2 [30] using GalaxyTrakr [21]. 

### 2.5. In Silico MLST and Serotyping 

The initial analysis and identification of the strains were performed using an in silico *Cronobacter* spp. MLST approach based on the information available at the MLST website (https://pubmlst.org/organisms/cronobacter-spp). The in silico *Cronobacter* serotype present in each sample was determined by batch screening in Ridom SeqSphere+ v9.0.8 (Ridom, Münster, Germany) using the genes described in Wang et al., 2021 [31] and available as part of the CroTrait pipeline (https://github.com/happywlu/CroTrait). 

### 2.6. Phylogenetic Relationship of the Strains by Whole Genome Multilocus Typing (Wgmlst) Analysis

The phylogenetic relationship of the strains was assessed by a whole genome multilocus sequence typing (wgMLST) analysis using Ridom SeqSphere+ v9.0.8, utilizing the contigs generated from the de novo assembly step for each overnight inoculated PIF sample. The genome of *C. zakazakii* ATCC BAA-894 (NC_009778.1) containing 4103 CDSs was used as a reference. The complete closed genomes of these *C. zakazakii* strains (NC_017933.1-ES15, NC_020260.1-SP291, NZ_CP011047.1-ATCC 29544, and NZ_CP012253.1-NCTC 8155) were used for comparison with the reference genome to establish a list of core and accessory genes. Genes that are repeated in more than one copy in any of the two genomes were removed from the analysis as failed genes. A task template then was created that contains both core and accessory genes for this reference *Cronobacter sakazakii* strain for any future testing (wgMLST scheme). Each individual locus from strain BAA-894 was assigned allele number 1. The assemblies for each individual *Cronobacter* closed genome in this study were queried against the task template, and if the locus was found and was different from the reference genome or any other queried genome already in the database, a new number was assigned to that locus and so on. After eliminating any loci that were missing from the genome of any strain used in the analyses, we performed the wgMLST analysis. These remaining loci were considered the core genome shared by the analyzed strains. Nei’s DNA distance method [32] was used for calculating the matrix of genetic distance, taking into consideration only the number of same/different alleles in the core genes. The resultant genetic distances were used to generate a Neighbor-Joining (NJ) tree. wgMLST uses the alleles number of each locus for determining the genetic distance and builds the phylogenetic tree. The use of allele numbers reduces the influence of recombination in the data set studied and allows for fast clustering determination of genomes.

## 3. Results

### 3.1. Cronobacter-Inoculated Sample-Enrichment Preparation for Nanopore Sequencing

*Cronobacter* grew in every enrichment culture for all PIF inoculated samples. The uninoculated PIF sample continued to test negative following enrichment. *Cronobacter* concentrations in each enrichment varied from 10^7^ to 10^9^ CFU/mL (Table 2). The BAM qPCR method detected the presence of *Cronobacter* in each overnight inoculated PIF enrichment. A schematic representation of the entire workflow as described in materials and methods is shown in Figure 1. All inoculated PIF samples resulted in positive qPCR signal at Cq values of 12.8 to 18.7 (Table 2). The enumeration results and qPCR Cq had an excellent correlation with an R2 of 0.93.

### 3.2. Nanopore Long-Read Sequencing Results

The DNA extracted from 1 mL of each of the 10 inoculated PIF overnight enrichment samples were sequenced using ONT for 48 h. The sequencing output for the run was 17.16 Gb in 3.25 million reads, of which 94% (3 M) passed the quality filter (minimum quality score of 10). Around 600 k reads were unclassified and were discarded. The read-length N50 was 8 kb, and there was enough data generated for each of the samples at different coverage levels to perform a successful genome assembly and sample characterization (Table 3). Reads below 4000 bp and quality scores below 10 were discarded for downstream analysis, resulting in 915,281 remaining reads. The estimated coverage per *Cronobacter* sample (genome size approximately 4.5 Mb) ranged from 55–420X. 

The preliminary *Cronobacter* species ID and the number of total reads matching each species per inoculated PIF sample were determined using Oxford Nanopore EPI2ME “What’s in my pot” (WIMP) workflow analysis (Table 2). Only one species was misclassified by WIMP as *C. universalis* (EB18) instead of the expected *C. turicensis*. According to the WIMP analysis, 90% to 99.6% of the classified reads per inoculated PIF sample belonged to the inoculated strain (Table 2). These numbers confirmed what was observed by qPCR for the same samples. According to these values, the concentration of each individual inoculated strain ranged from 1.5 × 10^7^ to 5.8 × 10^8^ CFU/mL. 

### 3.3. Nanopore Long-Read Genome Assembly of Cronobacter PIF-Enriched Samples

Since the observed *Cronobacter* spp. concentrations (Table 2) in inoculated PIF overnight enriched samples were similar to the concentrations observed for pure overnight cultures, the genome assembly was performed using the reads obtained for each sample, following the standard procedure for pure cultures [22,33]. The complete closed genome for each strain was de novo assembled using Flye v2.9 in a high-performance computing environment (Table 4). The assembly of the reads for each sample resulted in completely closed circular chromosomes and/or plasmids. The G+C mol% of these assemblies were 56.8% to 58.1%, which is within the range of the reported GC content for *Cronobacter* spp. strains [11]. The genome of some samples (E515 and E603) was composed only of a chromosome without the presence of any extra chromosomal elements. The remainder of the samples carried between two and three plasmids. Analysis of every contig for each sample with Kraken2 showed that all of them belonged to *Cronobacter* spp. The chromosome coverage for each sample varied from 65 to 516X. 

### 3.4. Multilocus Sequence Typing (MLST) and Serotyping Analysis

In silico MLST analysis showed that most of these strains belonged to eight known sequence types (STs) (ST1, 4, 8, 15, 19, 54, 80, and 81), with E791 belonging to a novel ST profile (151, 20, 111, 90, 195, 212, 239) (Table 5). Interestingly, the closest STs to E791 reported in the *Cronobacter* MLST database matched in four loci, and they all belonged to the species C. *dublinensis* (ST477, 479,576, and 662).

### 3.5. wgMLST Analysis of Cronobacter PIF Overnight Enriched Samples and Taxa Classification Using a Phylogenetic Tree

The phylogenetic relationship among the 37 *Cronobacter* spp. genomes (27 completely closed genome representative for each of the seven *Cronobacter* species publicly available at GenBank-Appendix A, and the 10 ONT assemblies from the inoculated overnight enriched PIF samples) was determined by a custom wgMLST analysis (Figure 2). A total of 3965 genes were used as templates for the analysis of the *Cronobacter* spp. strains. Every species was easily distinguished, and the wgMLST confirmed their current species definition. The wgMLST analysis grouped the 10 PIF inoculated samples according to their species and distinguished differences between strains from the same ST (e.g., ST8, differing in at least 34 alleles). This phylogenetic tree also confirmed that sample EB18 was indeed a *C. turicensis*.

## 4. Discussion

Considering the importance of powdered infant formula (PIF) safety, accurate detection and classification of *Cronobacter* spp. in PIF are crucial, particularly during outbreaks. Current methods for identification and confirmation include qPCR and extensive selective plating before whole genome sequencing (WGS) analysis. This time-consuming process requires nearly 2 weeks for isolate confirmation. The use of qPCR as a screening tool of the pre-enrichment broth and long-read sequencing analysis of qPCR-positive samples allows detection and full characterization of a *Cronobacter* strain in 3 to 4 days. While microbiological confirmation remains the standard, this new method can provide quicker insights to guide risk-management decisions during PIF outbreak scenarios, potentially shortening the time to identify the source of the outbreak and reduce illnesses risk by up to a week.

Since levels of *Cronobacter*, especially *C. sakazakii*, in overnight enrichments can reach 10^7^ to 10^9^ CFU/mL and *Cronobacter* makes up 90.0–99.6% of the culture (Table 3), samples from overnight enrichments can be treated as pure isolates and sequenced directly without the need of further selective enrichment or plating steps to isolate the *Cronobacter* strain from the other microbiota present in the overnight enrichment. Simple ONT sequencing and assembly using all reads has demonstrated in this limited study as adequate to completely close *Cronobacter* genomes. The data generated from each sample is manageable (compressed fastq files from 0.3 Gb to 1.8 Gb) and can produce a usable genome in about 0.5 to 2 h post-run in a high-performance environment. This reduces the overall analysis time compared to traditional culture methods and WGS, allowing detection of related strains in a few days (3 days from enrichment to usable data), compared to approximately 2 weeks (Figure 1).

Given the high relative populations of *Cronobacter* in each enrichment (90.0–99.6%) (Table 3), short-read sequencing could satisfactorily identify and characterize these strains, but a complete genome would not be recovered. Short reads would capture most of the genome, but some fragments and repetitive regions might be missed. In contrast, ONT allows for closing the complete genome (chromosome and/or plasmids) (Table 4). ONT also enables positive identification of most *Cronobacter* species in the PIF sample within just 1 h using WIMP, except for *C. turicensis*. This highlights a problem with accurate taxa identification using curated databases (In this case, RefSeq Database created on 8 June 2021). If the reference genome defining a species is missing in the database, proper identification may not be achieved. This problem can be mitigated by regular updating of databases improving their accuracy and comprehensiveness. Without these regular updates, these databases might lack essential information, leading to incomplete or incorrect identification of microorganisms. 

The observed STs in this study agreed with the expected results for the inoculated taxa per PIF samples (Table 1 and Table 5). The *Cronobacter* MLST database currently holds 4150 strains, with varying numbers for different STs (e.g., 499 strains belong to ST1 (reported as *C. sakazakii*), 615 strains to ST4 (reported as *C. sakazakii*), 125 strains to ST8 (reported as *C. sakazakii*), one strain to ST15 (reported as *C. sakazakii*), seven strains to ST19 (reported as *C. turicensis*), eight strains to ST54 (reported as *C. universalis*), seven strains to ST80 (reported as *C. dublinensis*), and 13 strains to ST81 (reported as *C. muytjensii*). In silico serotype analysis also matched expected results, showing high diversity of serotypes among the samples and within the same species (e.g., two different O types for *C. sakazakii* strains). ONT rapidly differentiated them using in silico MLST, which is crucial since most *Cronobacter sakazakii* causing more than 90% of infant illnesses belong to ST4 [13], including recent cases linked to PIF and breast pump equipment caused by ST4 strains [7]. 

It is important to use qPCR for early screening and quantification to determine the approximate concentration of *Cronobacter* in PIF enriched samples, which helps estimate the number of samples per flow cell for ONT sequencing. A previous study of ONT sequencing for Culture-Independent Diagnostics Tests (CIDT) of STECs in agricultural water overnight enrichments determined that a minimum of 10^6^ CFU/mL of STEC in a complex mixture was needed for successful genome closure using a single flow cell [1]. This study showed that when the target organism reached concentrations above 10^6^ CFU/mL and 90–99.6% of the sample, up to 10 *Cronobacter* genomes could be successfully closed with high coverage (100–300X). This suggests that more samples could be included, reducing the cost per sample. At current prices, the entire run costs around $860 (flow cell—$700, Sequencing kit library—$110, and DNA extraction—$50), with a cost of $86 USD per sample. Bulk purchasing of flow cells and running 24 samples instead of 10 could reduce the price per sample to around $25. The *Cronobacter* ONT method from PIF enriched samples has an advantage over STEC enrichments, as a positive PIF sample usually results in a single organism, reducing sample complexity [1].

This pilot study aimed to test nanopore sequencing for preliminary identification and characterization (MLST, serotyping, and wgMLST) within approximately 3 days from sample collection, comparing results to reference genomes (Table 1 and Figure 2). Sequencing with the new R10.4.1 flow cell showed higher accuracy than R9.4.1, as observed by many others [1,17,34]. Strains with previous sequences in NCBI clustered tightly together, with varied numbers of loci differences (Appendix A and Figure 2). For instance, sample E477 (ST8) differed in at least three loci from ATCC 29544 (ST1). However, issues such as indels and missing genes highlight the need for cautious interpretation and robust validation of ONT data compared to other sequencing methods like Illumina.

Currently, WGS in bacterial-related diseases focuses on sequencing microorganisms isolated from selective culture plates [7,35,36,37]. Clinical and diagnostic labs are transitioning to Culture Independent Diagnostic Tests (CIDTs) for rapid identification, but CIDTs do not yield physical isolates, hindering outbreak investigations by public health agencies. This in turn prolongs outbreak resolution [38]. Similarly, isolating foodborne bacteria for surveillance or case investigations, such as *Cronobacter*, takes around 1 week per isolate before WGS can be performed. There is a push for rapid, cost-effective, and highly accurate culture-independent methods for both clinical and food investigations [1,3,38,39]. Previous metagenomic approaches yielded closed STEC genomes but were not suitable for SNP-based source tracking [1]. While this study utilizes an example with *E. coli* in agricultural samples to illustrate the assembly limits with Oxford Nanopore Technologies (ONT) metagenomics, the principle of requiring high bacterial concentrations in enriched samples is broadly applicable. Specifically, achieving a sufficient bacterial load is essential to ensure reliable genome recovery across diverse sample types [1]. In both agricultural and powdered infant formula samples, high bacterial levels in the enrichment significantly (10^6^ to 10^9^ CFU/mL in the overnight enrichment) enhance the precision and completeness of ONT sequencing results, enabling more accurate pathogen identification and characterization. This study using newer ONT chemistry (R10.4.1) for *Cronobacter* identification and subtyping directly from overnight PIF enrichments shows higher quality than earlier results obtained for STEC using the R9.4.1 flow cells. Each inoculated strain (MLST and serotype) was accurately identified and subtyped nearly to the strain level (wgMLST). Similar results have been observed by other authors for certain strains of *Listeria monocytogenes* and STEC [25] and *Staphylococcus* [26], showing the improvement of ONT sequencing for outbreak investigations. However, protocols for nanopore sequencing, DNA extraction, and library preparation still vary, and validation for repeatability, reproducibility, and robustness is needed before implementation. 

## 5. Conclusions

In conclusion, utilizing ONT for *Cronobacter* identification and preliminary characterization from PIF samples can speed up finding the correct strain during outbreak investigations to approximately 3 days, in contrast to the nearly 2 weeks required by the current FDA methods. Despite high concentrations of *Cronobacter* in the inoculated overnight PIF enrichment, exceeding 10^7^ CFU/mL, a combined approach involving qPCR and nanopore sequencing is advocated in time-sensitive situations to enhance source tracking. qPCR can detect *Cronobacter* and estimate the concentration in the sample, predicting the adequacy of subsequent nanopore sequencing for complete genome recovery. Additionally, qPCR data can inform the number of samples feasible for sequencing in a single flow cell, reducing costs per sample. Furthermore, other samples can be included in the sequencing run when fewer samples are available. Moreover, when using this approach, regularly updating the databases used for taxa identification is essential to improve their accuracy and comprehensiveness, thus avoiding the misidentification of some species. 

## Figures and Tables

**Figure 1 microorganisms-12-02389-f001:**
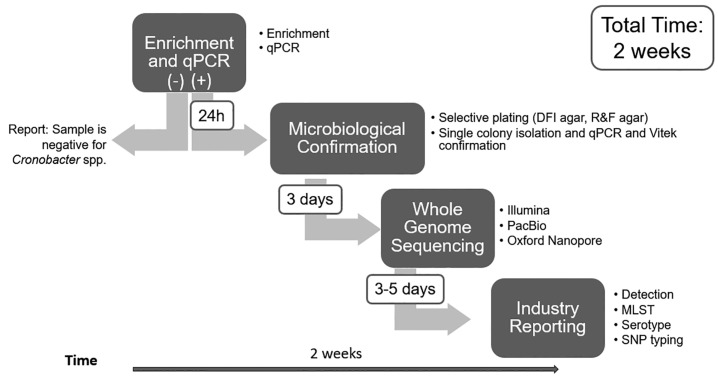
Flow diagram for detection, isolation, and full WGS characterization of *Cronobacter* spp. strains from PIF in Chapter 29 in the BAM.

**Figure 2 microorganisms-12-02389-f002:**
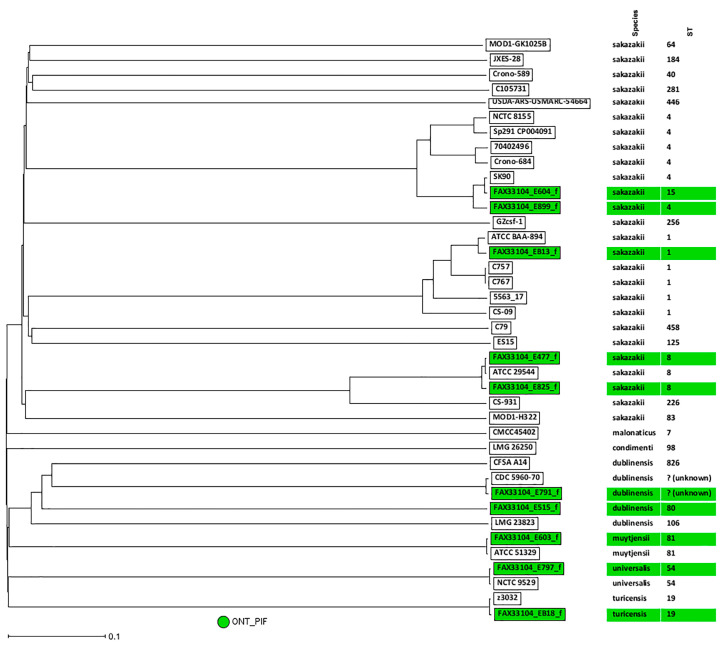
Neighbor joining (NJ) phylogenetic tree based on gene differences on the 3965 analyzed genes (allele based) of the 10 *Cronobacter* spp. overnight inoculated PIF genomes generated in this study with the other 27 *Cronobacter* spp. genomes, representing the seven *Cronobacter* described species (Appendix A). Highlighted in green are the 10 samples recovered from the overnight enrichment of inoculated PIFs.

**Table 1 microorganisms-12-02389-t001:** Metadata of the strains used in the spiking experiment in this study.

Samples	Cronobacter Species	Source	Country of Origin	Available at NCBI
E477	*sakazakii*	Human (throat)	Unknown	ATCC 29544
E515	*dublinensis*	Water	Switzerland	NA
E603	*muytjensii*	Unknown	unknown	ATCC 51329
E604	*sakazakii*	Human	Canada	SK90
E791	*dublinensis*	Human (blood)	USA	CDC 5960-70
E797	*universalis*	Water	UK	NCTC 9529
E825	*sakazakii*	Human (breast abscess)	USA	NA
EB13	*sakazakii*	Neonate (meningitis)	Switzerland	NA
EB18	*turicensis*	Neonate (meningitis)	Switzerland	NA
E899	*sakazakii*	Clinical	USA	NA

NA—not available.

**Table 2 microorganisms-12-02389-t002:** Taxonomic classification of the inoculated PIF samples and qPCR values obtained from the overnight enrichments.

Samples	*Cronobacter* Taxa Expected	*Cronobacter* Taxa Observed WIMP ^a^	*Cronobacter* Taxa Observed by Phylogenetic Tree	qPCR C_T_ Value	Estimated *Cronobacter* CFU/mL ^b^	*Cronobacter* ONT % DNA Sample
E477	*sakazakii*	*sakazakii*	*sakazakii*	13.6	1.9 × 10^8^	97.0
E515	*dublinensis*	*dublinensis*	*dublinensis*	17.0	1.5 × 10^7^	90.0
E603	*muytjensii*	*muytjensii*	*muytjensii*	13.1	2.8 × 10^8^	99.0
E604	*sakazakii*	*sakazakii*	*sakazakii*	12.1	5.8 × 10^8^	90.0
E791	*dublinensis*	*dublinensis*	*dublinensis*	13.4	2.2 × 10^8^	92.0
E797	*universalis*	*universalis*	*universalis*	12.4	4.3 × 10^8^	99.6
E825	*sakazakii*	*sakazakii*	*sakazakii*	12.7	3.7 × 10^8^	97.1
EB13	*sakazakii*	*sakazakii*	*sakazakii*	12.2	5.3 × 10^8^	98.2
EB18	*turicensis*	*universalis*	*turicensis*	12.4	4.4 × 10^8^	90.4
E899	*sakazakii*	*sakazakii*	*sakazakii*	15.3	5.3 × 10^7^	95.4

^a^ WIMP (“What’s in my pot” workflow in Epi2me) identified sample EB18 as composed by three different taxa: *universalis* (59%), *sakazakii* (20.4%), and *malonaticus* (11%). ^b^ in the overnight enrichment.

**Table 3 microorganisms-12-02389-t003:** Nanopore sequencing run statistics.

Samples	Total Reads	Total Mb	Estimated Coverage *Cronobacter* Genome All Reads (X)	Reads above 4000 bp	Total Mb above 4000 bp	Estimated Coverage *Cronobacter* Genome > 4 kb Reads (X)
E477	228,000	1069	238	82,998	732	163
E515	70,096	354	79	26,066	248	55
E603	523,730	2665	592	203,038	1889	420
E604	127,684	643	143	47,322	455	101
E791	257,927	1148	255	85,301	752	167
E797	278,012	1382	307	105,162	969	215
E825	210,799	1010	224	77,642	697	155
EB13	239,315	1228	273	91,575	870	193
EB18	319,323	1648	366	125,660	1168	260
E899	191,946	941	209	70,517	647	144

**Table 4 microorganisms-12-02389-t004:** Assembly statistics for each PIF inoculated sample in this study.

Samples	Contig No.	%GC Content	Genome Size (bp)	Genome Coverage (X)
E477	3	56.6	4,507,829; 93,905; 53,449	136; 156; 287
E515	1	57.9	4,487,108	65
E603	1	57.7	4,305,928	516
E604	3	56.6	4,412,859; 117,865; 52,143	115; 123; 179
E791	2	58.1	4,349,860; 166,041	208; 239
E797	2	57.9	4,075,540; 129,777	273; 306
E825	3	56.8	4,257,543; 97,419; 53,456	185; 163; 260
EB13	3	56.7	4,347,023; 131,190; 31,203	214; 265; 1269
EB18	4	57.2	4,384,296; 144,804; 53,716; 44,722	283; 357; 552; 375
E899	2	56.7	4,340,415; 53,472	176; 284

**Table 5 microorganisms-12-02389-t005:** MLST and serotyping of the inoculated PIF assembled samples.

Samples	ST ^a^	*Cronobacter* Taxa by ST	Serotype ^b^
E477	8	*sakazakii*	SO1
E515	80	*dublinensis*	DO2
E603	81	*muytjensii*	MuO2
E604	15	*sakazakii*	SO2
E791	novel	*dublinensis*	DO1a
E797	54	*universalis*	UO1
E825	8	*sakazakii*	SO1
EB13	1	*sakazakii*	SO1
EB18	19	*turicensis*	TO1
E899	4	*sakazakii*	SO2

^a^ https://pubmlst.org/organisms/cronobacter-spp; ^b^ https://github.com/happywlu/CroTrait.

## Data Availability

The raw ONT data for each individual sample were deposited in the GenBank under the BioProject accession number PRJNA1117853.

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
