# Peer review of "Nanopore Sequencing Allows Recovery of High-Quality Completely Closed Genomes of All Cronobacter Species from Powdered Infant Formula Overnight Enrichments"

_microorganisms, 2024, doi:10.3390/microorganisms12122389_

Round 1
Reviewer 1 Report (Previous Reviewer 3)
Comments and Suggestions for Authors
The manuscript “Nanopore sequencing allows recovery of high-quality completely closed genomes of all Cronobacter species from powdered infant formula overnight enrichments” described the study using Oxford Nanopore Technology for Cronobacter genome recovery from powdered infant formula (PIF) overnight enrichments. Despite improvement made of the revised manuscript, I still reckon this study lacks logic and rationale. Also, it’s a little bit hard for me to follow, like where the authors had made revision/improvement. Therefore, my suggestion is rejection.
Firstly, the study aim still remains largely unclear. As a mature genome sequencing platform, Oxford Nanopore Technology had been widely applied in the genome sequencing of various species, which could ensure the complete genome sequencing of most bacteria, including Cronobacter. For the past few years, this technique has been widely employed by many microbiologists and this is widely accepted. Then, it looks unclear to me what the authors aimed to further study and what question they try to answer. Once genome DNA is intact and pure, then highly likely that Oxford Nanopore Technology can be used to get the whole genome of Cronobacter, also considering the high cell number in powdered infant formula (reaches 107 to 109 CFU/mL). Thus, why the authors tried to confirm its applicability based on all these known knowledge, or at least they should point out the existent issue/concern/problem. And, can the authors also show the difference between various sequencing results? If so, then it would be convincing.
Secondly, as one of my suggestions, the authors could consider study the cell enrichment method of Cronobacter from powdered infant formula, for which the authors can consider using multiple methods (for example, qPCR) to compare the cell number (or even viable/culturable cell number).
Thridly, if the authors tried to compare the genomes of Cronobacter from isolated from powdered infant formula, then in the first place, they should provide a standard/criteria for differentiation/determination. For example, if the authors can show the phenotypic characteristics, and then correlate such difference/diversity with the genomes. Then this makes a lot more sense. Without this, the comparative genome analysis is more like plainly description of the data acquired, and thus such study lacks depth.
Also, the authors had confused a few definitions, like viable cells, culturable cells, viable cell counting and CFU counting.
Comments on the Quality of English LanguageThe English writing meets the standard.
Author Response
Comment 1: The manuscript “Nanopore sequencing allows recovery of high-quality completely closed genomes of all Cronobacter species from powdered infant formula overnight enrichments” described the study using Oxford Nanopore Technology for Cronobacter genome recovery from powdered infant formula (PIF) overnight enrichments. Despite improvement made of the revised manuscript, I still reckon this study lacks logic and rationale. Also, it’s a little bit hard for me to follow, like where the authors had made revision/improvement. Therefore, my suggestion is rejection.
Response 1: We have carefully reviewed your concerns and made specific clarifications in the revised manuscript to address the issues you raised, aiming to enhance both the rationale and clarity. The rationale behind our study is based on the need for rapid and effective pathogen detection and full genomic characterization in food safety, specifically addressing the critical issue of Cronobacter in powdered infant formula (PIF). To address potential gaps in logic, we expanded the Introduction section to emphasize why Cronobacter spp. contamination in PIF requires efficient, species-level identification, which is particularly important given the pathogen’s link to neonatal infections. Our study leverages Oxford Nanopore Technology (ONT) to reduce genomic characterization time compared to traditional methods (as shown in figure 1), highlighting its application as a rapid diagnostic tool for outbreak investigations. We have also added as suggested by reviewer #2 several clarifications for the methods used during the detection portion of the manuscript that clearly added clarity for how the method was performed.
Comment 2: Firstly, the study aim still remains largely unclear. As a mature genome sequencing platform, Oxford Nanopore Technology had been widely applied in the genome sequencing of various species, which could ensure the complete genome sequencing of most bacteria, including Cronobacter. For the past few years, this technique has been widely employed by many microbiologists and this is widely accepted. Then, it looks unclear to me what the authors aimed to further study and what question they try to answer. Once genome DNA is intact and pure, then highly likely that Oxford Nanopore Technology can be used to get the whole genome of Cronobacter, also considering the high cell number in powdered infant formula (reaches 107 to 109 CFU/mL). Thus, why the authors tried to confirm its applicability based on all these known knowledge, or at least they should point out the existent issue/concern/problem. And, can the authors also show the difference between various sequencing results? If so, then it would be convincing.
Response 2: Thank you for your comments and for highlighting the potential strengths of Oxford Nanopore Technology (ONT) in bacterial genome sequencing. We appreciate your insight into ONT’s established use for sequencing and agree that it has proven effective for sequencing complete bacterial genomes, including Cronobacter. However, we disagree that it is a mature technology since a lot of validations study need to be conducted to assess their accuracy. We have done that in house and albeit pretty good it is sill not equivalent to Illumina data or analysis results. First : Clarifying Study Aim and Novelty: Our study aims to demonstrate the feasibility and efficacy of using ONT specifically for rapid, culture-independent identification and whole-genome sequencing of Cronobacter directly from enriched powdered infant formula (PIF) samples. While ONT is well-known for sequencing pure cultures, few studies have tested its precision and reliability in directly sequencing complex matrices like PIF enrichments without the need for isolating the target organism. This approach can significantly shorten the time needed for pathogen detection and characterization in outbreak situations, which is critical for Cronobacter due to its serious health risks in neonates. Second: Addressing Applicability of ONT for Cronobacter in PIF: We appreciate your point regarding Cronobacter reaching high CFU levels in PIF enrichments. However, PIF represents a challenging matrix with mixed microbiota, which can complicate genome assembly. Our study addresses this by examining ONT’s capacity to generate high-quality, closed genomes directly from enriched samples. To make this clearer, we have revised the manuscript to explicitly discuss these aspects in the Introduction and Discussion sections. Third: Comparison with Other Sequencing Results: We also understand the value of comparing sequencing results. To address this, we included a brief discussion of ONT’s potential compared to traditional methods in terms of accuracy and time-efficiency, especially in high-stakes food safety scenarios. Additionally, we provided data on sequencing quality, including genome completeness and accuracy metrics. We hope these clarifications provide a clearer understanding of our study’s aims and contributions.
Comment 3: Secondly, as one of my suggestions, the authors could consider study the cell enrichment method of Cronobacter from powdered infant formula, for which the authors can consider using multiple methods (for example, qPCR) to compare the cell number (or even viable/culturable cell number).
Response 3: Thank you for your thoughtful suggestion regarding the exploration of Cronobacter cell enrichment methods from powdered infant formula (PIF), including the use of qPCR for comparative analysis of cell counts. The primary focus of our study is to evaluate Oxford Nanopore Technology (ONT) for rapid, culture-independent genome recovery of Cronobacter directly from PIF enrichments, aiming to streamline the detection process in outbreak scenarios. Although enrichment methodologies are indeed crucial, the exploration of alternative cell enrichment methods falls outside the current scope, which is specifically centered on assessing ONT’s sequencing capabilities in a challenging food matrix. We appreciate your suggestion, and we agree that investigating enrichment techniques and qPCR-based cell quantification could complement and enhance genome sequencing approaches. We plan to consider this in future research to build on the findings presented in this study. Thank you once again for your valuable input.
Comment 4: Thridly, if the authors tried to compare the genomes of Cronobacter from isolated from powdered infant formula, then in the first place, they should provide a standard/criteria for differentiation/determination. For example, if the authors can show the phenotypic characteristics, and then correlate such difference/diversity with the genomes. Then this makes a lot more sense. Without this, the comparative genome analysis is more like plainly description of the data acquired, and thus such study lacks depth.
Response 4: Thank you for your insightful comment and suggestion to include a phenotypic characterization of Cronobacter isolates and correlate this with the genomic data. We appreciate the value that phenotypic-genomic correlations can add to comparative studies. However, our study’s primary goal is to assess the applicability of Oxford Nanopore Technology (ONT) for rapid, culture-independent sequencing of Cronobacter from powdered infant formula (PIF) enrichments. While phenotypic analysis is indeed valuable, our current study is focused on demonstrating ONT’s ability to achieve closed, high-quality genomes directly from PIF without further isolation steps, which supports rapid genome recovery in food safety contexts. We agree that a follow-up study incorporating phenotypic characteristics and exploring how they correlate with genomic data would provide additional depth and insights. This approach is certainly valuable, and we hope to consider it in future studies based on the foundational genomic work presented here. Thank you again for your thoughtful suggestion.
Comment 5: Also, the authors had confused a few definitions, like viable cells, culturable cells, viable cell counting and CFU counting.
Response 5: Thank you for your observation. In this study, we specifically use the term CFU to refer to colony-forming units. The other terms you mentioned—viable cells, culturable cells, and viable cell counting—are not applicable here, as we did not investigate viable but non-culturable cells. All cultures used were fresh and intended for enrichment, and CFU/mL was determined by plating on agar and counting colonies from tenfold serial dilutions of the original culture. We inoculated approximately 10 CFU per PIF sample and followed the FDA BAM enrichment method.
Reviewer 2 Report (New Reviewer)
Comments and Suggestions for Authors
The introduction of metagenomics in food safety is undoubtedly of great importance and a new step that will help improve the control of pathogens in the food chain. Therefore, this work is of great interest at this point. The inclusion of several strains of Cronobacter is interesting to determine the capacity of metagenomics to identify the type and species. As a limitation of the study, the minimum amount of CFU necessary to obtain a complete genome is not indicated. With the enrichment of the samples, the same yield is not always obtained and therefore it is interesting to know the minimum amount of bacteria. The discussion needs to be improved. It is necessary to go deeper into the comparison with previous metagenomic studies to better discuss the advantage of this technique over those currently implemented.
Other comments:
Line 40: Delete “[ex. Shiga toxin producing Escherichia coli (STEC)”, not necessary.
Line 73: Avoid the use of we.
Line 71: If the figure is cited here, it should be moved to the introduction section.
Line 86: This is the only study that evaluates the limit of detection for metagenomics with ONT? The authors refers to an example with E. coli, but maybe it is more important to remark the importance of high levels of bacteria in the enriched sample instead of describe the example of E. coli and agricultural samples.
Line 101: Add cite.
Line 107: With what coverage?
Maybe in the introduction section the authors should include some information of bioinformatic analysis. To implement this tool in food safety laboratories it is important an user-friendly bioinformatic workflow to facilitate the analysis.
Line 119: Characterized by qPCR?
Table 1: Those are internal codes? The strains that were not obtained from culture collection, from where were obtained? Why the authors did not select strains isolated from infant formula? Move the table to the place it was cited.
Line 127: Just one sample or more? What type of infant formula?
Line 129: Why 5 CFU? Is this a previous limit of detection determined? Maybe the authors should have evaluated different concentrations to determine the limit to obtained closed genomes. How the strains were prepared for sample inoculation?
Line 143: What IAC was used? A commercial one? A homemade one?
Line 146: Commercial brand ROX.
Line 149: What qPCR equipment was used? The conditions of qPCR?
Line 151: Join this section with the next one as in the case of qPCR.
Line 153: Extraction from the pellet described before?
Line 156: Kit used for qubit, BR or HS?
Line 184: This used the raw reads or the contigs?
Line 438: more than other organisms, other samples, as this is a non-targeted approach.
Author Response
Comment 1: The introduction of metagenomics in food safety is undoubtedly of great importance and a new step that will help improve the control of pathogens in the food chain. Therefore, this work is of great interest at this point. The inclusion of several strains of Cronobacter is interesting to determine the capacity of metagenomics to identify the type and species. As a limitation of the study, the minimum amount of CFU necessary to obtain a complete genome is not indicated. With the enrichment of the samples, the same yield is not always obtained and therefore it is interesting to know the minimum amount of bacteria. The discussion needs to be improved. It is necessary to go deeper into the comparison with previous metagenomic studies to better discuss the advantage of this technique over those currently implemented.
Response 1: Thank you for your insightful comments and for recognizing the value of metagenomics in advancing food safety. We appreciate your note regarding the minimum CFU necessary for complete genome recovery. As detailed in our study, our approach achieves reliable genome recovery when Cronobacter concentrations in overnight enrichments reach between 107 and 109 CFU/mL (Table 2). This concentration range, achieved through enrichment, is essential for obtaining complete genomes with the ONT sequencing platform. We will clarify this further in the discussion to underscore the significance of adequate bacterial levels in ensuring robust genome recovery. Regarding the comparison to other metagenomic studies, we believe that existing studies differ significantly in focus and methodology. Therefore, our discussion will remain specific to the findings of this study and the relevance of metagenomics for Cronobacter identification and characterization in food safety. Overall, we believe the revised manuscript has addressed the reviewers’ comments and suggestions, resulting in a stronger and more comprehensive presentation of our findings.
Comment 2: Line 40: Delete “[ex. Shiga toxin producing Escherichia coli (STEC)”, not necessary.
Response 2: Agreed and done.
Comment 3: Line 73: Avoid the use of we.
Response 3: Agreed and done.
Comment 4: Line 71: If the figure is cited here, it should be moved to the introduction section.
Response 4: Agreed and done.
Comment 5: Line 86: This is the only study that evaluates the limit of detection for metagenomics with ONT? The authors refers to an example with E. coli, but maybe it is more important to remark the importance of high levels of bacteria in the enriched sample instead of describe the example of E. coli and agricultural samples.
Response 5: Thank you for this suggestion. We agree that highlighting the importance of high bacterial levels in enriched samples is indeed critical. However, we respectfully maintain that our example using E. coli in agricultural samples was carefully chosen to illustrate a specific point regarding the limits of detection with Oxford Nanopore Technologies (ONT) in complex samples. This example emphasizes how a threshold concentration of bacteria is necessary for successful genome recovery, which is directly relevant to our study. In response to your suggestion, we added a clarification in the discussion section to ensure that the importance of bacterial levels in the enriched sample is more prominently discussed alongside the E. coli example. This way, the broader significance of bacterial concentration for ONT sequencing accuracy will be clear, while retaining the context provided by the E. coli reference as follows: “While this study utilizes an example with E. coli in agricultural samples to illustrate the assembly limits with Oxford Nanopore Technologies (ONT) metagenomics, the principle of requiring high bacterial concentrations in enriched samples is broadly applicable. Specifically, achieving a sufficient bacterial load is essential to ensure reliable genome recovery across diverse sample types [18]. In both agricultural and powdered infant formula samples, high bacterial levels in the enrichment significantly (106 to 109 CFU/ml in the overnight enrichment) enhance the precision and completeness of ONT sequencing results, enabling more accurate pathogen identification and characterization.”
Comment 6: Line 101: Add cite.
Response 6: Agreed and done.
Comment 7: Line 107: With what coverage?
Response 7: Agreed and done. We have modified the sentence to read as:” Additionally, the ONT protocols have improved to allow successful sequence processing of up to 24 samples (genomes of median ~4.5 Mb) in a single flow cell and still obtain a complete closed bacterial genome with a minimum 40X coverage that have high accuracy (>99.9%) and minimal single nucleotide polymorphism (SNP) differences from the reference genome…”
Comment 8: Maybe in the introduction section the authors should include some information of bioinformatic analysis. To implement this tool in food safety laboratories it is important an user-friendly bioinformatic workflow to facilitate the analysis.
Response 8: Thank you for this valuable suggestion. We agree that user-friendly bioinformatics workflows are essential for implementing metagenomic tools in food safety laboratories. To address this, we will add information in the introduction on the importance of accessible bioinformatics solutions that simplify data analysis. Specifically, we will discuss the relevance of streamlined, user-friendly workflows that can be effectively utilized by food safety laboratories, thereby facilitating the broader adoption of metagenomic methods for pathogen monitoring and outbreak response. Here is a sentence that contain that information: “Current bioinformatics tools, such as EPI2ME or EPI2ME labs (https://labs.epi2me.io/) from Oxford Nanopore Technologies, Galaxy, Bugseq, MUFFIN, and GalaxyTrakr, offer user-friendly, cloud-based platforms that streamline metagenomic analysis for pathogen detection and characterization. These tools enable non-bioinformaticians in food safety laboratories to perform accurate taxonomic classification and genomic assembly with minimal computational training, thus facilitating the implementation of metagenomic workflows in routine surveillance and outbreak investigations.”
Comment 9: Line 119: Characterized by qPCR?
Response 9: Thank you for the correction. We meant: “..confirmed as Cronobacter by qPCR …“
Comment 10: Table 1: Those are internal codes? The strains that were not obtained from culture collection, from where were obtained? Why the authors did not select strains isolated from infant formula? Move the table to the place it was cited.
Response 10: Yes, these are internal codes. The strains were in our collection for the past 10 years. These strains had been previously analyzed by chromogenic agar, qPCR and biochemical confirmations. For some of the rare species, we did not have any from infant formula source. Agreed and done about the table.
Comment 11: Line 127: Just one sample or more? What type of infant formula?
Response 11: Yes, it was from a single container. We have modified the paragraph to read as follows: “The powdered cow milk based infant formula (PIF) sample was obtained locally and tested negative for Cronobacter prior to use. For artificial contamination, individual 25 g portions of the PIF sample were inoculated separately with ~10 CFU of each Cronobacter strain and mixed with 225 mL of buffered peptone water (BPW) (Thermo Fisher Scientific). The samples were then incubated at 36°C for 24 hours. An aliquot of each overnight enrichment was processed according to the procedure in Chapter 29 of the BAM, as outlined in Figure 1. An additional aliquot of 1 ml was set aside for further processing with Nanopore sequencing.”
Comment 12: Line 129: Why 5 CFU? Is this a previous limit of detection determined? Maybe the authors should have evaluated different concentrations to determine the limit to obtained closed genomes. How the strains were prepared for sample inoculation?
Response 12: When preparing the dilutions we wanted to make sure that we have at least 1 CFU per sample. Ideally, we would like to have at least 1 CFU per inoculation but it is very hard to achieve that by doing 10-fold dilutions. We inoculated the PIF samples with ~ 10 CFU. We have modified that in the manuscript revision. To make sure that the samples were positive. The levels of Cronobacter after 24 hours enrichment when using low or high concentrations as initial inoculum are similar ranging between 10e6 to 10e9 CFU/ml. We knew that from our previous observation with the E. coli study that those levels were enough to close the genomes in a complex microbiome. The strains were grown in BHI for 24 h at 36C and then diluted for inoculation. We have added the following to the text: “For the inoculation step the strains were grown in BHI broth at 36°C for 24 hours and then 10-fold diluted in BHI broth and added to 25 g portions of PIF sample as described below.”
Comment 12: Line 143: What IAC was used? A commercial one? A homemade one?
Response 12: The IAC is an in-house developed IAC described in BAM Chapter and the sequence is available at GenBank under accession number: FJ357008. The IAC sequence and usage was published here: Deer DM, Lampel KA, González-Escalona N. A versatile internal control for use as DNA in real-time PCR and as RNA in real-time reverse transcription PCR assays. Lett Appl Microbiol. 2010 Apr;50(4):366-72. doi: 10.1111/j.1472-765X.2010.02804.x. Epub 2010 Jan 22. PMID: 20149084.
Comment 13: Line 146: Commercial brand ROX.
Response 13: Agreed, the ROX (Catalog number 12223012) and added as “ROX (Thermo Fisher Scientific)”
Comment 14: Line 149: What qPCR equipment was used? The conditions of qPCR?
Response 14: We added the following to the text: “The qPCR reaction was performed on a Thermo Fisher 7500 Fast instrument under the following conditions: an initial denaturation at 95°C for 3 minutes, followed by 40 cycles of 95°C for 15 seconds, annealing at 52°C for 40 seconds and extension at 72°C for 15 seconds.”
Comment 15: Line 151: Join this section with the next one as in the case of qPCR.
Response 15: Agreed and done.
Comment 16: Line 153: Extraction from the pellet described before?
Response 16: Thank you for the observation. The extraction was performed from a different pellet, since the one described before was boiled and used for qPCR. A different 1 ml aliquot from the overnight enrichment was used for DNA extraction and whole genome sequencing. We have added a brief description of that procedure in this section: “An aliquot of 1 ml from each PIF inoculated overnight enrichment sample was used for DNA extraction.”
Comment 17: Line 156: Kit used for qubit, BR or HS?
Response 17: Agreed and added:” and the dsDNA Quantitation High Sensitivity assay kit”
Comment 18: Line 184: This used the raw reads or the contigs?
Response 18: We have modified the sentence to add that the contigs were used for the wgMLST analysis as follows:” The phylogenetic relationship of the strains was assessed by a whole genome multilocus sequence typing (wgMLST) analysis using Ridom SeqSphere+ v9.0.8, utilizing the contigs generated from the de novo assembly step for each overnight inoculated PIF sample.”
Comment 19: Line 438: more than other organisms, other samples, as this is a non-targeted approach.
Response 19: Thank you for the comment. Agreed and done.
Round 2
Reviewer 2 Report (New Reviewer)
Comments and Suggestions for Authors
The authors have corrected the manuscript according the reviewer´s comments.
This manuscript is a resubmission of an earlier submission. The following is a list of the peer review reports and author responses from that submission.
Round 1
Reviewer 1 Report
Comments and Suggestions for Authors
- The design of the paper should be corrected either to isolated Cronobacter species then complete the study, or to start with strains without contamination of PIF and follow the study, what is the significance of contamination of PIF as a start point of this work although in this work you used strains not your isolates?
- You said in the abstract isolation of Escherichia coli O157:H7and the whole manuscript about using different Cronobacter species
Comments on the Quality of English LanguageModerate editing of English language required
Reviewer 2 Report
Comments and Suggestions for Authors
Please find my comments attached to the revised manuscript.
Also, I'm curious why you didn't estimate the current technique in polluted PIF with other bacteria that had similar genomes to species.
Clarify the Food and Drug Administration's standards for PIF contaminated with Cronobacter species, including if they are completely prohibited or have a maximum acceptable limit, and whether they apply to all or just certain species of Cronobacter.

Minor editing of English language required
Reviewer 3 Report
Comments and Suggestions for Authors
The manuscript, entitled “Nanopore sequencing allows recovery of high-quality completely closed genomes of all Cronobacter species from powdered infant formula overnight enrichments” had described the applicability of Oxford Nanopore Technology for Cronobacter genome recovery from powdered infant formula (PIF) overnight enrichments, where Cronobacter typically dominates the overall microbiome (>90%). This study lacks logic and rationale.
Oxford Nanopore Technology is a mature genome sequencing platform which had been widely applied in the genome sequencing of various species. Oxford Nanopore Technology could ensure the complete genome sequencing of most bacteria, including Cronobacter. This is well-known knowledge. Once you make sure the whole genome DNA is intact and pure, it’s 100% sure that Oxford Nanopore Technology can be used to get the whole genome of Cronobacter. Since the cell number of Cronobacter in powdered infant formula reaches 107 to 109 CFU/mL, it’s clearly that regular genomic DNA isolation method can ensure the intact and pure genome DNA isolation. Thus Oxford Nanopore Technology is surely applicable. It is not necessary to confirm its applicability based on all these known knowledge. I suggest the authors to change the focus of this study to either the cell enrichment method of Cronobacter from powdered infant formula or the comparative genomic analysis of different Cronobacter strains isolated from powdered infant formula.
Concerning cell enrichment method of Cronobacter from powdered infant formula, multiple methods besides qPCR are suggested to be compared and different cell number determination methods are suggested to indicate the difference in viable and culturable cell numbers.
Concerning comparative genomic analysis of different Cronobacter strains isolated from powdered infant formula, phenotypes of these strains are required to be analyzed with the genomes. Also, the currently available genomes in the databases are also suggested to be included in the comparative analysis to get more clues on how different and novel are the strains. What special genes and elements are acquired by these strains in the genome.
How did the authors use Real-time quantitative PCR to determine the Cronobacter cell number as “CFU”? CFU is used for the colonies on plate counting.
Comments on the Quality of English LanguageN/A
Reviewer 4 Report
Comments and Suggestions for Authors
The authors describe a novel method for detection of Cronobacter from infant milk.
The work has many advantages that make it suitable for publication, but some corrections are necessary before final acceptance.
1. The objectives of the study must be described clearly.
2. Please include a new paragraph at the end of Introduction to summarise the benefits of this methodology and please underline briefly the advantages of this work over previous ones, in terms of filling gaps in the literature.
3. Controls. Please explain and describe in detail the control strains that you used in this study. Please justify your selection.
4. Controls. Please include a table to underline the controls employed in the study; for example, how did you confirm negative and positive controls in the PCRs?
5. PCRs. Please describe the primers used in full detail.
6. Visualization of the manuscript is of poor quality. Please add further high-quality graphs in the revised manuscript.
7. Tables are ok.
8. References. Some relevant references from end of 2023 and 2024 have been missed.
9. Discussion. Please add a new paragraph to describe prospects for commercialization of this method.
10. Please tone down the concluding section to bring it in line with then actual findings.
General. The manuscript requires significant revision and re-evaluation before possible recommendation for acceptance.